# The Efficacy of Fluvoxamine in Anxiety Disorders and Obsessive-Compulsive Disorder: An Overview of Systematic Reviews and Meta-Analyses

**DOI:** 10.3390/ph18030353

**Published:** 2025-02-28

**Authors:** Michel Haddad, Luiz Henrique Junqueira Dieckmann, Thiago Wendt Viola, Melissa Ribeiro de Araújo, Naielly Rodrigues da Silva, Jair de Jesus Mari

**Affiliations:** 1Department of Psychiatry, Brazilian Clinical Research Institute, São Paulo 01404-000, Brazil; haddad@grupobipp.com.br (M.H.); dieckmann@grupobipp.com.br (L.H.J.D.); melrib.araujo@gmail.com (M.R.d.A.); naiellyrodrigues@grupobipp.com.br (N.R.d.S.); 2Brain Institute of Rio Grande do Sul, School of Medicine, Pontifical Catholic University of Rio Grande do Sul (PUCRS), Porto Alegre 90610-000, Brazil; thiago.wendt@pucrs.br; 3Department of Psychiatry, Universidade Federal de São Paulo, São Paulo 04017-030, Brazil

**Keywords:** fluvoxamine, panic disorder, social anxiety disorder, obsessive-compulsive disorder, systematic reviews

## Abstract

**Objective**: This systematic review aims to evaluate the efficacy of fluvoxamine in the treatment of anxiety disorders and obsessive-compulsive disorder (OCD) by synthesizing evidence from systematic reviews and meta-analyses. **Methods**: We conducted a literature search in PubMed and the Cochrane Central Register of Controlled Trials, focusing on fluvoxamine’s efficacy in generalized anxiety disorder (GAD), social anxiety disorder (SAD), panic disorder (PD), and OCD. We included systematic reviews and meta-analyses of randomized controlled trials (RCTs) comparing fluvoxamine to a placebo or other drugs. The quality of evidence from the included reviews was assessed using A Measurement Tool to Assess Systematic Reviews—version 2 (AMSTAR-2). **Results**: The study included fourteen systematic reviews (five for OCD, three for SAD, and six for PD), covering thirty-seven RCTs (sixteen for OCD, six for SAD, and fifteen for PD), with a total of 3621 patients (1745 with OCD, 1034 with SAD, and 842 with PD). A high-quality systematic review demonstrated that fluvoxamine is superior to a placebo in improving symptoms and the response rates for OCD. Three meta-analyses comparing fluvoxamine to clomipramine in OCD found no significant differences in efficacy regarding symptom improvement. Two additional systematic reviews, both rated as high quality, confirmed the superiority of fluvoxamine in reducing symptom severity and improving the response rates in patients with SAD compared to a placebo. However, the findings for PD were inconsistent. A meta-analysis, also rated as high quality, found that while fluvoxamine showed better response rates than a placebo, the difference was not statistically significant. **Conclusions**: Overall, the efficacy of fluvoxamine in the treatment of OCD and SAD was demonstrated. While some reviews highlighted its potential in alleviating GAD, its impact on panic-specific outcomes remained inconsistent.

## 1. Introduction

Anxiety disorders are a common group of mental disorders with a lifetime prevalence estimated at 33.7% [1,2]. This group encompasses a range of conditions, including Generalized Anxiety Disorder (GAD), Social Anxiety Disorder (SAD), Panic Disorder (PD), and others [1]. Although obsessive-compulsive disorder (OCD) shares some of the symptoms observed in anxiety disorders, the Diagnostic and Statistical Manual of Mental Disorders, 5th version (DSM-5) does not classify it as an anxiety disorder. Instead, it is categorized as an obsessive-compulsive and related disorder [3]. Both anxiety disorders and OCD are associated with significant impairments in social, occupational, and academic functioning, as well as a reduced quality of life [4,5,6].

Anxiety and depression, though often co-occurring and sharing some overlapping symptoms and treatment options, are distinct psychiatric conditions with unique characteristics [7,8,9]. Anxiety is primarily characterized by excessive fear, worry, and related behavioral disturbances [3,10]. These symptoms are future oriented and often involve a heightened perception of threat [10]. Depression is primarily marked by anhedonia, low mood/irritability, and other symptoms such as feelings of worthlessness, fatigue, and changes in sleep or appetite [3]. In general, the symptoms of depression are often past oriented and linked to feelings of loss or hopelessness [11].

The treatment of anxiety disorders and OCD has evolved significantly over the past few decades, with pharmacotherapy playing a pivotal role. Among the available treatments, Selective Serotonin Reuptake Inhibitors (SSRIs) have become the first-line option for both anxiety disorders and OCD, due to their efficacy and relatively favorable side effect profile [12]. Fluvoxamine, one of the earliest SSRIs developed, has been shown to effectively treat OCD symptoms since the 1980s [13]. Additionally, its anxiolytic effects have gained increasing recognition, making it a promising treatment for various anxiety disorders [14].

Evidence strongly supports the critical role of sigma-1 receptor agonism in the mechanism of action of fluvoxamine, which exhibits the highest affinity for sigma-1 receptors among SSRIs and binds to these receptors in the human brain at therapeutic doses [15,16]. These receptors influence neurotransmitter systems, intracellular proteostasis, calcium homeostasis, and the dynamic excitatory/inhibitory balance in the brain which have been linked to neuroprotective, anxiolytic, and antidepressant properties [16,17,18].

Meta-analyses and systematic reviews have been crucial in elucidating fluvoxamine’s therapeutic benefits, particularly its effectiveness in reducing anxiety and obsessive-compulsive symptoms [19]. Furthermore, fluvoxamine has shown a favorable side effect profile, particularly in comparison to other SSRIs and Selective Norepinephrine Reuptake Inhibitors (SNRIs), making it a viable option for the long-term management of these conditions [20]. However, there is still a lack of information as to whether fluvoxamine, with its unique mechanism of action, could perform superiorly in relation to other pharmacological treatments, particularly newer SSRIs that were developed after fluvoxamine’s approval.

Given the growing body of evidence, this manuscript provides an overview of the efficacy of fluvoxamine in the treatment of anxiety disorders and OCD. By synthesizing the findings from systematic reviews and meta-analyses, we aim to offer a comprehensive evaluation of fluvoxamine’s efficacy in the management of these disorders. Also, by reviewing its efficacy in comparison to alternative treatments, our study aims to assess whether fluvoxamine presents superior efficacy for anxiety and obsessive-compulsive disorders.

## 2. Materials and Methods

This study was conducted following the Cochrane Method for Overviews of Reviews, and we adhered to the Preferred Reporting Items for Systematic Reviews and Meta-Analyses (PRISMA) 2020 guidelines [21,22,23]. We searched the following electronic bibliographic databases: MEDLINE (PubMed) and the Cochrane Central Register of Controlled Trials database. There were no restrictions on language or publication year. The search strategies included the following general terms: fluvoxamine AND (“panic disorde*” OR anxiety OR “social phobia” OR “generalized anxiety disorder” OR “obsessive-compulsive disorde*” OR OCD) AND (“systematic review” OR “meta-analysis”). We included systematic reviews and meta-analyses of randomized controlled trials (open-label, single-blind, or double-blind) that assessed the efficacy of fluvoxamine in treating GAD, SAD, PD, or OCD, comparing it to a placebo or other drugs. The participants were adults diagnosed with psychiatric disorders, based on the criteria outlined in the Diagnostic and Statistical Manual of Mental Disorders (DSM) or the International Classification of Diseases (ICD), who had received treatment with fluvoxamine, a placebo, or other drugs.

The exclusion criteria were as follows: (1) non-systematic literature reviews; (2) reviews lacking standardized diagnostic confirmation using the DSM or ICD criteria, such as studies on Research Domain Criteria (RDoC); (3) reviews exclusively focused on specific populations (e.g., children, adolescents, the elderly, pregnant women, menopausal women, perinatal anxiety, or postpartum women); (4) reviews focused on the efficacy and use of fluvoxamine in COVID-19 patients; (5) reviews of studies that included populations with other psychiatric disorders or comorbidities (e.g., substance use disorders, neurocognitive disorders, psychotic disorders, personality disorders, or suicide); (6) reviews focusing solely on the adverse effects of fluvoxamine or other antidepressants; (7) reviews exclusively targeting dose-finding studies of antidepressants; (8) reviews comparing fluvoxamine to psychotherapeutic interventions; (9) reviews focused on treatment guidelines; (10) reviews that did not specifically evaluate the efficacy of fluvoxamine; (11) reviews including preclinical trials; (12) reviews focused solely on the economic evaluation of medications; and (13) review protocols.

The study selection process was conducted in two stages: a screening phase (title and abstract reading) and an eligibility phase (full-text reading). Two investigators (JJM and TWV) independently carried out these stages; in cases of discrepancies, a third analyst (NRS) was consulted to reach a consensus. Following the selection process, four independent analysts (JJM, MH, LHJD, and MRA) extracted data, and an extraction table was constructed, incorporating data from head-to-head comparisons between fluvoxamine, a placebo, and other drugs. To assess the inter-rater reliability, a Cohen’s kappa score was estimated to evaluate the agreement on the study eligibility between the two primary investigators, resulting in a high level of concordance (κ = 0.889). We extracted the following information: first author’s name, year of publication, identification of the randomized clinical trials (RCTs) included in the review, the number of individuals included in the analysis, main findings, and, when available, type of meta-analysis (pairwise or network meta-analysis [NMA]) and meta-analytical estimates (effect sizes and measures of dispersion intervals). Effect size measures included the odds ratio (OR), relative risk (RR), mean difference (MD), standardized mean difference (SMD), and Hedge’s g statistic, that measure the effect size for the difference between means [24]. As measures of dispersion, we considered 95% confidence intervals (95% CI) or 95% credible intervals (95% CrI). Additionally, we extracted heterogeneity findings from the meta-analysis that indicated these analyses (Appendix A).

We aimed to extract individual data from the RCTs included in each review. If individual data were not available, pooled estimates were extracted instead. Additionally, we also conducted an additional search using the same search strategy as in Medline, but excluding the last combination of terms (“systematic review” OR “meta-analysis”). This additional search was performed aiming to identify individual RCTs not included in the currently available systematic reviews. It was limited to the last year of inclusion of individual studies in the most recent systematic review about each psychiatric disorder. To calculate the effect size for these individual studies, we determined the proportion of patients who responded to each treatment arm and then computed the effect size as the difference between these proportions. We also calculated the standard error of this difference to estimate the precision of the effect size, using a formula that accounts for variability in each group’s response rate. Finally, we estimated the 95% confidence interval (CI) for the effect size.

The quality of evidence from the included reviews was assessed using A Measurement Tool to Assess Systematic Reviews—version 2 (AMSTAR-2), a practical critical appraisal tool specifically designed for rapid assessments of the quality of systematic reviews of RCTs [25]. Each review was evaluated based on all 16 items of the tool (Appendix A). A review was considered of high quality if it scored points on all 16 items. Reviews that missed points on only two criteria—reporting the review protocol or presenting the funding information for all the included studies, were also considered of high quality. This adjustment acknowledges that these practices have become more common in recent years, and some included reviews that were published more than 15 years ago. Reviews that missed more points but scored higher than 10 on the AMSTAR-2 were categorized as of moderate quality. Reviews with a score below 10 were considered of low quality due to multiple items missing points according to the tool. One investigator (TWV) applied this tool, and the results were discussed with three additional investigators (JJM, MH, and NRS).

To calculate the Corrected Covered Area (CCA) for assessing the overlap of primary studies included in the systematic reviews, we first identified all primary studies across the reviews and created a matrix to record their presence or absence. The CCA was then computed by considering the extent of overlap between the studies, adjusting for the total number of studies and reviews to avoid the inflation of overlap due to the size of the reviews. This study was registered with PROSPERO (International prospective register of systematic reviews), number CRD42024557845.

## 3. Results

Fourteen systematic reviews were included in our analysis (Figure 1). Of these, five focused on OCD [26,27,28,29,30], three on SAD [31,32,33], and six on PD [34,35,36,37,38,39], covering a total of 35 RCTs. Additionally, two individual RCTs were included in the OCD analysis [40,41], bringing the total number of RCTs analyzed to thirty-seven (Appendix A). The average duration of treatment in the included RCTs was 10.5 weeks for OCD, 13.7 weeks for SAD, and 7.5 weeks for PD. After applying the exclusion criteria, no systematic reviews or meta-analyses on the effect of fluvoxamine on GAD were included.

### 3.1. Obsessive-Compulsive Disorder

Five systematic reviews and meta-analyses were identified, involving a total of 16 RCTs and 1745 participants, focusing on comparisons between fluvoxamine and a placebo, clomipramine, desipramine, and sertraline. The detailed results and characteristics of these reviews on the efficacy of fluvoxamine for treating OCD are summarized in Table 1.

Five systematic reviews and meta-analysis along with one independent RCT compared fluvoxamine versus a placebo [26,27,28,29,30,40], with only one systematic review rated as of high quality according to the AMSTAR-2 criteria [30]. These studies comprised findings of eight RCTs [13,42,43,44,45,46,47,48], exhibiting a very high overlap of RCTs (CCA = 80%).

Several studies comparing fluvoxamine to a placebo reported significant efficacy in treating OCD symptoms, as measured by using the Yale–Brown Obsessive-Compulsive Scale (Y-BOCS) and other relevant scales. Notably, the only high-quality systematic review, as assessed by using AMSTAR-2, found that fluvoxamine led to greater symptom reduction across all the measured outcomes compared to a placebo [30]. The sole exception was a study from Piccinelli (1995), which did not show statistical superiority of fluvoxamine over a placebo in symptom improvement, although this study was rated as low quality by using AMSTAR-2 [26].

Three meta-analyses compared fluvoxamine to clomipramine, all of which were rated as low quality by using AMSTAR-2 [26,28,29]. These studies included data from five RCTs [49,50,51,52,53], showing a low overlap of RCTs (CCA = 20%). All the studies reported non-significant differences in the efficacy between fluvoxamine and clomipramine of improving OCD symptoms.

A comparison between fluvoxamine and desipramine was reported in one meta-analysis, which also received a low AMSTAR-2 score [26]. This study, which included a single RCT [13], found fluvoxamine to be superior to desipramine in improving OCD symptoms, with a statistically significant effect size (Hedge’s g = 1.19 CI = 0.52, 1.86). Additionally, a recent RCT demonstrated that fluvoxamine was more effective than sertraline in reducing OCD symptoms, with the difference being statistically significant (ES = 0.32 CI = 0.08, 0.55) [41]. Overall, the findings suggest that fluvoxamine is generally effective in reducing OCD symptoms, but its superiority over other treatments, such as clomipramine and sertraline, is based on a limited number of clinical trials.

**Table 1 pharmaceuticals-18-00353-t001:** Characteristics of included systematic reviews and meta-analyses on the efficacy of fluvoxamine in the treatment of OCD.

Review	Source	Included RCTs	Time of Follow-Up (wk)	Blinding (RCTs)	No. of Patients per Arm of Treatment (Fluvoxamine, Comparator)	Measure of Efficacy	Summary Estimates (ES, 95% CI)	Main Findings	AMSTAR
Fluvoxamine vs. Placebo Primary studies overlap = High (CCA = 80%)									
Piccinelli 1995 [26]	MA	Goodman, 1989 [13]	6–8	db	21, 21	Symptom improvement assessed by using the Y-BOCS score	Hedge’s g = 0.57, (0.37, 0.77)	Non-significant difference	Low
		Jenike, 1990 [42]	10	db	18, 20				
		Rasmussen, 1992 [43]	10	db	160, 160				
	MA	Jenike, 1990 [42]	10	db	18, 20	Symptom improvement assessed by using the NIMH-OC score	Hedge’s g = 0.29 (0.07, 0.51)	Non-significant difference	
		Rasmussen, 1992 [43]	10	db	160, 160				
Greist 1995 [27]	MA	Greist, 1992 [47]	10	db	160, 160	Mean change in obsessive-compulsive disorder symptoms	ES = 0.50 (0.25, 0.75)	Fluvoxamine superior to a placebo	Low
		Rasmussen, 1992 [43]	10	db	160, 160				
Ackerman and Greenland 2002 [28]	MA	Goodman, 1989 [13]	6–8	db	21, 21	Symptom improvement assessed by using the Y-BOCS score	PD = −4.84 (−7.78, −1.83)	Fluvoxamine superior to a placebo	Low
		Jenike, 1990 [42]	10	db	18, 20				
		Mallya, 1992 [44]	10	db	14, 14				
		Goodman, 1996 [46]	10	db	78, 78				
Soomro 2008 [30]	SR	Goodman, 1989 [13]	6–8	db	21, 21	Symptom improvement assessed by using the Y-BOCS score	WMD = −3.87 (−5.69, −2.04)	Fluvoxamine superior to a placebo	High
		Hollander, 2002 [45]	12	db	117, 120				
		Jenike, 1990 [42]	10	db	18, 20				
		Nakajima, 1996 [48]	8	db	60, 33				
		Goodman, 1996 [46]	10	db	78, 78				
Soomro 2008 [30]	SR	Goodman, 1996 [46]	10	db	78, 78	Symptom improvement assessed by using the NIMH-OC score	WMD = −0.99 (−1.54, −0.44)	Fluvoxamine superior to a placebo	High
		Jenike, 1990 [42]	10	db	18, 20				
Soomro 2008 [30]	SR	Goodman 1996 [46]	10	db	78, 78	Symptom improvement assessed by using the NIMH-OC or CGI-I scales	SMD = −0.42 (−0.61, −0.23)	Fluvoxamine superior to a placebo	High
		Hollander 2002 [45]	12	db	117, 120				
		Jenike, 1990 [42]	10	db	18, 20				
Soomro 2008 [30]	SR	Goodman 1989 [13]	6–8	db	21, 21	Response rate	RR = 2.68 (1.58, 4.56)	Fluvoxamine superior to a placebo	High
		Goodman 1996 [46]	10	db	78, 78				
		Hollander 2002 [45]	12	db	117, 120				
		Nakajima, 1996 [48]	8	db	81, 44				
O’Connor 2006 [40]	RCT	-	20	db	11, 10	Symptom improvement assessed by using the Y-BOCS score	ES = 0.51	Fluvoxamine superior to a placebo	-
Fluvoxamine vs. Clomipramine Primary studies overlap = Low (CCA = 20%)									
Piccinelli 1995 [26]	MA	Smeraldi, 1992 [49]	12	db	5, 5	Improvement in obsessive-compulsive symptoms	Hedge’s g = −0.04 (−0.43, 0.35)	Non-significant difference	Low
		Freeman, 1994 [50]	10	db	34, 30				
	MA	Freeman 1994 [50]	10	db	34, 30	Improvement in obsessive symptoms	Hedge’s g = −0.15 (−0.64, 0.34)	Non-significant difference	
	MA	Freeman, 1994 [50]	10	db	34, 30	Improvement in compulsive symptoms	Hedge’s g = −0.07 (0.56, 0.42)	Non-significant difference	
Ackerman and Greenland 2002 [28]	MA	Smeraldi, 1992 [49]	12	db	5, 5	Symptom improvement assessed by using the Y-BOCS score	PD = 1.23 (−1.11, 3.56)	Non-significant difference	Low
		Freeman, 1994 [50]	10	db	34, 30				
		Koran, 1996 [51]	10	db	34, 39				
		Milanfranchi, 1997 [52]	9	db	13, 13				
		Mundo, 2001 [53]	10	db	115, 112	Symptom improvement assessed by using the Y-BOCS score	ES = 0.3 (−1.54, 2.14)	Non-significant difference	
Choi 2009 [29]	SR	Koran, 1996 [51]	10	db	34, 39	Symptom improvement assessed by using the Y-BOCS score	Fluvoxamine: ES = −7.7 Clomipramine: ES = −7.3	Non-significant difference	Low
		Mundo, 2001 [53]	10	db	115, 112	Symptom improvement assessed by using the Y-BOCS score	Fluvoxamine: ES = −12.3 Clomipramine: ES = −12.0	Non-significant difference	
Fluvoxamine vs. Desipramine No overlap									
Piccinelli 1995 [26]	MA	Goodman, 1990 [54]	8	db	21, 19	Improvement in obsessive-compulsive symptoms	Hedge’s g = 1.19 (0.52, 1.86)	Fluvoxamine superior to desipramine	Low
Fluvoxamine vs. Sertraline No overlap									
Brar 2022 [41]	RCT	-	12	ol	25, 25	Symptom improvement assessed by using the Y-BOCS score	ES = 0.32 (0.08, 0.55)	Fluvoxamine superior to sertraline	-

Note: Abbreviations: CI, confidence interval; CGI-I, Clinical Global Impression scale-Improvement; db, double blinded; ES, effect size; MA, meta-analysis; NIMH-OC, National Institute of Mental Health Obsessive Compulsive Rating Scale; PD, pooled difference; SMD, standardized mean differences; SR, systematic review; RR, risk ratio; WMD, weighted mean difference; Y-BOCS, Yale–Brown Obsessive-Compulsive Scale.

### 3.2. Social Anxiety Disorder

Three reviews [31,32,33] were included in our analysis, assessing the efficacy of fluvoxamine in SAD. These reviews involved six RCTs [55,56,57,58,59,60] and exhibited a very high overlap of RCTs (CCA = 92%). They focused exclusively on the direct comparisons of fluvoxamine versus a placebo and involved 1034 participants in total (Table 2).

Both high-quality systematic reviews as assessed by using AMSTAR-2 demonstrated fluvoxamine’s superiority in reducing SAD symptoms’ severity and achieving better response rates compared to a placebo [32,33]. Liu (2018) reported significant clinical improvement favoring fluvoxamine, including mean differences in the Liebowitz Social Anxiety Scale (LSAS) (MD = 11.90 CI = 8.09, 15.71), Clinical Global Impression Severity ratings (CGI-S; MD = 0.52 CI = 0.33, 0.72), and response rates (OR = 1.71 CI = 1.30, 2.24) [32]. Additionally, fluvoxamine was associated with reduced psychosocial impairment, as measured by using the Sheehan Disability Scale (SDS; MD = 2.11 CI = 1.03, 3.18) [32]. A network meta-analysis (NMA) conducted by Williams (2020) further confirmed the reduction in symptoms’ severity (MD = −2.12 CI = −21.88, −17.64) and improved response rates (OR = 1.89 CI = 1.14, 3.12) [33].

In addition, a systematic review by Hansen (2008) [31], which received a moderate score on the AMSTAR-2 assessment, found fluvoxamine to be significantly superior to a placebo in reducing SAD symptoms, as measured by using the LSAS (MD = −12.3 CI = −16.3; −8.22) [31]. However, while global clinical improvement also favored fluvoxamine, the difference was not statistically significant (OR = 1.49 CI = 0.94; 2.36) [31].

### 3.3. Panic Disorder

Six included reviews comprised two meta-analyses [34,37], two systematic reviews [35,36], and two NMA [38,39]. These reviews assessed the comparative efficacy of fluvoxamine in the treatment of PD, involving a total of 15 RCTs [61,62,63,64,65,66,67,68,69,70,71,72,73,74,75] and 842 participants. The reviews encompassed the results of both single-arm fluvoxamine treatment and direct comparisons of fluvoxamine monotherapy with a placebo, imipramine, maprotiline, brofaromine, sertraline, clomipramine, and inositol (Table 3).

Five reviews compared fluvoxamine with a placebo in the treatment of PD [34,35,37,38,39]. These reviews incorporated the findings from nine RCTs [61,62,63,66,67,68,72,73,75] which exhibited a moderate overlap of primary studies (CCA = 77%). The results were inconsistent, and only one review was rated as of high quality according to the AMSTAR-2 assessment [39]. This Cochrane NMA found fluvoxamine to be superior to a placebo in terms of response at the end of treatment, although the difference was not statistically significant (RR = 0.86, CI = 0.53 to 1.05) [39]. However, fluvoxamine significantly outperformed a placebo in achieving the remission of PD (RR = 0.77, CI = 0.50 to 0.95) [39]. No clear advantages were found for fluvoxamine in reducing the panic attack frequency (MD = 0.06 CI = −3.46, 3.55), panic severity (SMD = −0.17 CI = −0.79, 0.45), or agoraphobic symptoms (SMD = −0.50 CI = −1.42, 0.41) [39]. Additionally, dropout rates were similar between the fluvoxamine and placebo groups (RR = 1.17 CI = 0.85 to 1.66) [39].

Two moderate-quality reviews, as assessed by using AMSTAR-2, also demonstrated the effect of fluvoxamine on PD compared to a placebo [37,38]. The superiority of fluvoxamine in improving anxiety symptoms was measured by using various scales, such as the Clinical Anxiety Scale (CAS) (Hedge’s g fluvoxamine: 2.064 ± 0.61; placebo: 0.619 ± 0.8) [67] and (Hedge’s g fluvoxamine: 1.641 ± 0.79; placebo: 0.656 ± 0.89) [63], Hamilton Rating Scale for Anxiety (HAMA) (Hedge’s g fluvoxamine: 1.98 ± 0.9; placebo: 1.02 ± 1.04) [73], and the State Trait Anxiety Inventory (STAI) (Hedge’s g fluvoxamine: 1.82 ± 1.63; placebo: 0.29 ± 2.62) [37,66]. However, no significant effect of fluvoxamine was observed on the Panic-Associated Symptoms Scale (PAAS) (Hedge’s g; fluvoxamine: 0.638 ± 3.3; placebo: 0.572 ± 25) [37,63]. In the network meta-analysis conducted by Du and colleagues (2021), fluvoxamine was found to be marginally superior to a placebo, but not statistically significantly (OR 1.21, CI = 0.96–1.53) [38].

Controversial results were found in two low-quality reviews, as assessed by using AMSTAR-2, comparing fluvoxamine to a placebo [34,35]. Boyer and colleagues (1995) reported a significant improvement in PD symptoms with fluvoxamine [34]. However, in Mochcovitch et al.’s (2010) systematic review [35], one RCT included indicated fluvoxamine’s superiority [68], while another RCT found fluvoxamine to be inferior to a placebo in terms of the percentage of panic-free patients at the end of the treatment [61]. The efficacy of fluvoxamine was also demonstrated in a single-arm study, showing a significant reduction in anxiety symptoms, as measured by using the STAI and Zung Self-Rating Anxiety Scale (ZUNG) (Hedge’s g = 1.03 ± 2.15) [71].

Two meta-analyses, considered low-quality systematic reviews according to the AMSTAR-2, compared fluvoxamine with imipramine in regard to PD treatment [35,36], pooling data from two RCTs [68,70] with no overlap of primary studies as detailed in Table 3. One review found no difference between fluvoxamine and imipramine in response to treatment or reduction in functional impairment and fear symptoms [36]. The other review, however, found that fluvoxamine was inferior to imipramine in regard to the percentage of patients panic-free at the endpoint [35].

In comparison with other drugs, fluvoxamine was found to be superior to maprotiline (Hedge’s g 1.850 ± 1.03) and brofaromine (Hedge’s g 1.890 ± 1.05) in reducing anxiety symptoms as assessed by using the HAMA scale. These findings were reported in a systematic review with a meta-analysis [37], which included one RCT comparing fluvoxamine to maprotiline [65] and another comparing it to brofaromine [74]. One low-quality review, according to AMSTAR-2, indicated that sertraline and clomipramine were significantly superior to fluvoxamine when the response to treatment as assessed by using PASS was measured, but no significant differences were found in regard to the reduction in functional impairment or fear symptoms as assessed by using SDS or the fear questionnaire (FQ) when comparing fluvoxamine to either sertraline or clomipramine [36]. This review also highlighted that there was no significant difference in the response to treatment with fluvoxamine compared to inositol based on the data from one RCT [36,69].

## 4. Discussion

The present overview of systematic reviews provides a comprehensive assessment of the efficacy of fluvoxamine in the treatment of anxiety disorders and OCD. The results indicate that fluvoxamine is effective in treating both OCD and SAD, but its effectiveness in relation to PD is less consistent. In OCD, fluvoxamine demonstrated significant reductions in symptoms compared to a placebo, as confirmed by high-quality systematic reviews. However, its efficacy compared to clomipramine does not show significant differences, as observed in some clinical trials. In SAD, fluvoxamine consistently outperformed a placebo in reducing the symptom severity and improving social functioning, as demonstrated in multiple systematic reviews. Conversely, the evidence for fluvoxamine’s effectiveness in relation to PD is mixed. While some reviews reported positive outcomes in reducing general anxiety symptoms in PD patients, others found its effects to be limited or not significantly different from a placebo.

Anxiety disorders and OCD are prevalent and debilitating psychiatric conditions, and SSRIs such as fluvoxamine have been widely studied as first-line pharmacological interventions. As an SSRI, fluvoxamine increases serotonin levels, which aims to improve mood and reduce anxiety symptoms. Additionally, fluvoxamine has unique pharmacodynamic properties that differentiate it from other SSRIs. For example, it exhibits a high affinity for sigma-1 receptors, which are involved in the modulation of various neurotransmitter systems, including dopamine and glutamate [18].

The activation of sigma-1 receptors has been shown to have anxiolytic and neuroprotective effects, suggesting that fluvoxamine’s action on these receptors may contribute to its therapeutic effects in relation to anxiety and OCD [18,76,77]. Furthermore, fluvoxamine’s sigma-1 receptor activity may explain its relatively favorable tolerability profile, as sigma-1 agonism has been associated with reduced neurotoxicity and fewer adverse effects [78].

In OCD, fluvoxamine has shown substantial efficacy in reducing the symptom severity compared to a placebo. Five systematic reviews and meta-analyses, including those by Soomro et al. (2008) and Ackerman and Greenland (2002), consistently reported improvements in OCD symptoms, as measured by using the Y-BOCS and other relevant instruments [26,27,28,29,30]. Notably, the high-quality systematic review by Soomro (2008) demonstrated significant reductions in OCD symptoms, further confirming the utility of fluvoxamine in managing this condition [30]. However, the lack of statistical superiority in Piccinelli’s (1995) study suggests that fluvoxamine’s effects may not be universal across all populations and study designs [26]. Despite these variations, fluvoxamine generally outperformed a placebo, confirming its role as a key pharmacological option for treating OCD.

When compared to clomipramine, fluvoxamine’s efficacy in OCD was less clear, with three meta-analyses showing no significant differences between the two drugs [26,28,29]. Clomipramine, a tricyclic antidepressant, has long been regarded as the gold standard for OCD treatment, suggesting that while fluvoxamine is effective, it may not surpass clomipramine in its efficacy. However, fluvoxamine’s more favorable side effect profile, as highlighted in the earlier literature, may promote a more tolerable long-term option for some patients [20]. Furthermore, the findings from a recent randomized controlled trial showing fluvoxamine’s superiority over sertraline in symptom improvement reinforce its efficacy and suggest that fluvoxamine may be a viable alternative to other SSRIs [41].

In SAD, fluvoxamine demonstrated robust efficacy, particularly in reducing the symptom severity and improving psychosocial functioning. Liu (2018) and Williams (2020), both high-quality systematic reviews, reported significant reductions in the symptom severity, as measured by using the LSAS and the CGI ratings [32,33]. These findings align with the previous literature suggesting that fluvoxamine is an effective treatment for SAD [57]. However, Hansen’s (2008) review [31], which reported non-significant global improvements despite symptom reductions, underscores the complexity of treating SAD and suggests that further research is needed to clarify the contexts in which fluvoxamine is most effective [31]. This variability also reflects the heterogeneous nature of SAD, where patients may exhibit varying responses to SSRIs depending on the symptom severity, duration of illness, and comorbid conditions.

It is noteworthy that several reviews and guidelines recommend fluvoxamine as a first-line treatment for SAD [79,80]. Fluvoxamine and Fluvoxamine CR received a level 1 grade for the treatment of SAD in the Canadian clinical practice guidelines [81]. Further supporting these findings, Pelissolo et al. (2019) reviewed the therapeutic strategies for SAD and confirmed that SSRIs, including fluvoxamine, are effective first-line treatments [82]. Moreover, a systematic review including Japanese patients also found SSRIs to be a valid option for the pharmacotherapy of SAD, including fluvoxamine [83]. While current evidence supports the use of fluvoxamine for treating SAD, further high-quality RCTs with larger sample sizes and longer follow-up periods are needed to comprehensively confirm its efficacy and safety profile. Thus, clinicians should weigh the benefits against the potential adverse effects when considering fluvoxamine as a treatment option for SAD.

In the context of PD, fluvoxamine’s efficacy is less conclusive. Several reviews, including those by Guaiana (2023) and Andrisano (2013), reported mixed results regarding fluvoxamine’s effectiveness in reducing panic symptoms and agoraphobia [37,39]. While some studies, such as the Cochrane network meta-analysis by Guaiana (2023), demonstrated a significant remission of panic symptoms with fluvoxamine, other outcomes, such as panic attacks frequency and reductions in agoraphobic symptoms, did not show significant improvements compared to a placebo. Specifically, the analysis indicated that fluvoxamine was more effective than a placebo in achieving treatment response and remission outcomes, although it ranked lower than other SSRIs, such as paroxetine and venlafaxine [39]. These findings suggest that, while fluvoxamine may be effective in some cases, its efficacy in treating PD may be limited compared to its success in relation to other anxiety disorders. Furthermore, the moderate overlap of primary studies in regard to PD reviews (CCA = 66%) highlights the need for more robust, large-scale studies to confirm these findings. The heterogeneous outcomes found in Mochcovitch et al. (2010) and Boyer (1995) suggest that fluvoxamine may not be as effective as other SSRIs or tricyclic antidepressants like imipramine, particularly in reducing the frequency of panic attacks and achieving a full remission [34,35].

Despite these limitations, fluvoxamine has shown promise in PD treatment, particularly in reducing general anxiety symptoms. The CAS and HAMA scales indicated significant reductions in anxiety symptoms across multiple studies [37,67]. Additionally, Du (2021) demonstrated that fluvoxamine was marginally superior to a placebo in treating PD, though the differences were not statistically significant [38]. These findings suggest that while fluvoxamine may not be the most robust option for PD, it remains a viable treatment for certain patient populations, particularly those who may not respond well to other SSRIs or who have comorbid anxiety disorders.

Fluvoxamine, as an SSRI, exhibits a delayed onset for therapeutic effects, which may initially lead to an exacerbation of anxiety symptoms, potentially affecting patient adherence and satisfaction during the early phases of treatment [84]. A recommended clinical approach involves initiating treatment with low doses and combining fluvoxamine with benzodiazepines to mitigate the side effects and minimize the “activation” of the effects commonly associated with SSRIs [84]. While the current body of evidence observed in our analysis provides moderate support for fluvoxamine as an effective treatment for panic disorder, its efficacy does not surpass that of other SSRIs routinely used as first-line treatments.

The main limitations of this study are as follows. The search for systematic reviews was limited to two primary databases (PubMed and the Cochrane Database of Systematic Reviews). Also, we did not find any systematic reviews that included data on fluvoxamine for GAD. We note that the disparities in the number of studies for each disorder may bias the results, with less representation for SAD and PD compared to the extensively studied OCD. To minimize omissions, we reviewed the references of the retrieved studies and the main reviews available in the literature. There was no need to contact the study authors, as no additional statistical analyses were performed on the presented data. The extent of overlap among the primary studies was reported for each review, and we adhered to standardized review methods, providing a detailed description of the inclusion and exclusion criteria of the studies analyzed. Regarding the format of this review as an overview of systematic reviews, unlike an umbrella review, we did not perform new meta-analytic calculations to synthesize the data across systematic reviews. While our overview of systematic reviews provides a comprehensive synthesis and critical evaluation of the existing evidence, an umbrella review could have offered additional summary information through statistical aggregation, potentially providing further insights.

Overall, the findings from this overview highlight fluvoxamine’s efficacy across various anxiety disorders and OCD, with particularly strong evidence supporting its use in OCD and SAD. Fluvoxamine’s ability to reduce symptom severity and improve psychosocial functioning in these disorders aligns with its established role as a first-line SSRI. However, its efficacy in PD is less consistent, and future research should aim to clarify the contexts in which fluvoxamine may be most effective, particularly in comparison to other pharmacological treatments. Nevertheless, clinicians should consider the specific needs and characteristics of each patient when selecting fluvoxamine, particularly in regard to PD, where alternative treatments may offer more consistent outcomes.

## Figures and Tables

**Figure 1 pharmaceuticals-18-00353-f001:**
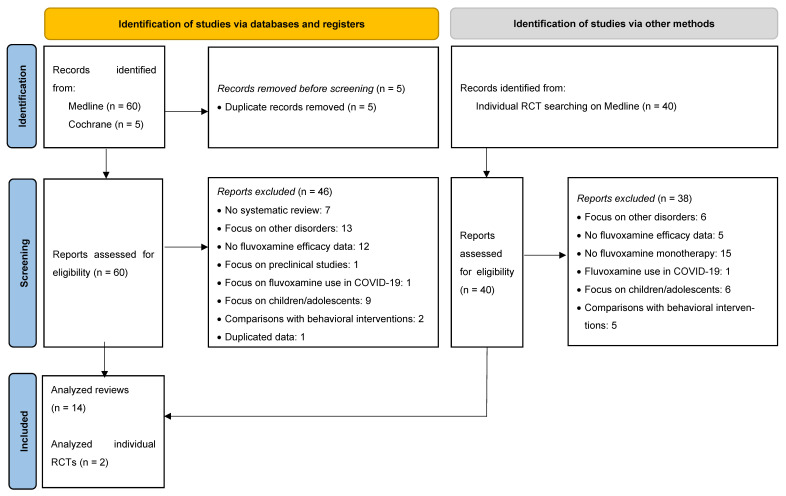
PRISMA 2020 flow diagram for articles included in the manuscript [22].

**Table 2 pharmaceuticals-18-00353-t002:** Characteristics of included systematic reviews and meta-analyses on the efficacy of fluvoxamine in the treatment of social anxiety disorder.

Review	Source	Included RCTs	Time of Follow-Up (wk)	Blinding (RCTs)	No. of Patients per Arm of Treatment (Fluvoxamine, Comparator)	Measure of Efficacy	Summary Estimates (ES, 95% CI)	Main Findings	AMSTAR
Fluvoxamine vs. Placebo Primary studies overlap = Very High (CCA = 92%)									
Liu 2018 [32]	MA	Davidson, 2004 [55]	12	db	121–126	Symptom severity assessed by using LSAS	MD = 11.90 (8.09–15.71)	Fluvoxamine superior to a placebo	High
		Stein, 2003 [57]	24	db	56–53				
		Stein, 1999 [58]	12	db	42–44				
		Westenberg, 2004 [60]	12	db	146–148				
		Davidson, 2004 [55]	12	db	121–126	Symptom severity assessed by using CGI-S	MD = 0.52 (0.33–0.72)	Fluvoxamine superior to a placebo	
		Stein, 2003 [57]	24	db	56–53				
		Westenberg, 2004 [60]	12	db	146–148				
		Asakura, 2007 [56]	10	db	176–89	Response Rate	OR = 1.71 (1.30–2.24)	Fluvoxamine superior to a placebo	
		Davidson, 2004 [55]	12	db	121–126				
		Stein, 2003 [57]	24	db	56–53				
		Stein, 1999 [58]	12	db	42–44				
		Westenberg, 2004 [60]	12	db	146–148				
		Asakura, 2007 [56]	10	db	175–89	Psychosocial impairment assessed by using SDS	MD = 2.11 (1.03–3.18)	Fluvoxamine superior to a placebo	
		Davidson, 2004 [55]	12	db	121–126				
		Stein, 2003 [57]	24	db	55–52				
		Westenbeg, 2004 [60]	12	db	146–148				
Hansen 2008 [31]	MA	Davidson, 2004 [55]	12	db	121, 126	Clinical Response assessed by using LSAS	MD = −12.3 (−16.3– −8.22)	Fluvoxamine superior to a placebo	Moderate
		Stein, 1999 [58]	12	db	42, 44				
		Westenberg, 2004 [60]	12	db	146, 148				
		Davidson, 2004 [55]	12	db	121, 126	Clinical global impression of improvement	OR = 1.49 (0.94–2.36)	Fluvoxamine superior to a placebo, but not statistically significantly	
		Stein, 1999 [58]	12	db	42, 44				
		Westenberg 2004 [60]	12	db	146, 148				
Williams 2020 [33]	NMA	Asakura 2007 [56]	10	db	182, 89	1: Reduction in symptom severity 2: Response to treatment	1: MD = −2.12 (−21.88–17.64) 2: OR = 1.89 (1.14–3.12)	1: Fluvoxamine superior to a placebo 2: Fluvoxamine superior to a placebo	High
		Davidson 2004 [55]	12	db	121–126				
		Stein 1999 [58]	12	db	48, 44				
		Stein 2003 [57]	12	db	57, 55				
		Van Vliet 1994 [59]	12	db	15, 15				
		Westenberg 2004 [60]	12	db	149, 151				

Note: Abbreviations: CI, confidence interval; CGI-S, Clinical Global Impression Severity of Illness; db, double blinded; LSAS, Liebowitz Social Anxiety Scale; NMA, network meta-analysis; MA, meta-analysis; MD, mean difference; OR, odds ratio; SDS, Sheehan Disability Scale.

**Table 3 pharmaceuticals-18-00353-t003:** Characteristics of included systematic reviews and meta-analyses on the efficacy of fluvoxamine in the treatment of panic disorder.

Review	Source	Included RCTs	Time of Follow-Up (wk)	Blinding (RCTs)	No. of Patients per Arm of Treatment (Fluvoxamine, Comparator)	Measure of Efficacy	Summary Estimates (ES, 95% CI)	Main Findings	AMSTAR
Fluvoxamine vs. Placebo Primary studies overlap = Moderate (CCA = 77%)									
Guaiana 2023 [39]	NMA	Asnis 2001 [61]	8	db	93, 95	1: Response to treatment 2: Remission 3: Frequency of PD attacks 4: Reduction in PD symptoms 5: Agoraphobia symptoms	1: RR = 0.86 (0.53–1.05) 2: RR = 0.77 (0.50–0.95) 3: MD = 0.06 (−3.46–3.55) 4: SMD = −0.17 (−0.79–0.45) 5: SMD = −0.50 (−1.42–0.41)	1: No significant difference 2: Fluvoxamine significantly better than a placebo 3: No significant difference 4: No significant difference 5: No significant difference	High
		Black 1993 [63]	8	db	25, 25				
		Den Boer 1990 [66]	8	db	20 (fluvoxamine), 20 (ritanserin), 19 (placebo)				
		Hoehn-Saric 1993 [67]	8	db	25, 25				
		Nair 1996 [68]	8	db	50, 50				
		Sharp 1996 [73]	12	db	29, 28				
Andrisano 2012 [37]	MA	Hoehn-Saric 1993 [67]	8	db	18, 18	Reduction in anxiety symptoms assessed by using CAS	Hedge’s g Fluvoxamine: 2.064 ± 0.61 Placebo: 0.619 ± 0.8	Fluvoxamine superior to a placebo	Moderate
		Black 1993 [63]	8	db	23, 23	Reduction in anxiety symptoms assessed by using CAS	Hedge’s g Fluvoxamine: 1.641 ± 0.79 Placebo: 0.656 ± 0.89	Fluvoxamine superior to a placebo	
		Black 1993 [63]	8	db	23, 23	Reduction in PD symptoms assessed by using PASS	Hedge’s g Fluvoxamine: 0.638 ± 3.3 Placebo: 0.572 ± 25	No significant difference	
		Nair 1996 [68]	8	db	50, 50	Reduction in anxiety symptoms assessed by using CAS	Hedge’s g Fluvoxamine: 1.03 ± 0.55 Placebo: 0.87 ± 0.5	No significant difference	
		Sharp 1996 [73]	12	db	29, 28	Reduction in anxiety symptoms assessed by using HAMA	Hedge’s g Fluvoxamine: 1.98 ± 0.9 Placebo: 1.02 ± 1.04	Fluvoxamine superior to a placebo	
		Den Boer 1990 [66]	8	db	20 (fluvoxamine), 20 (ritanserin), 19 (placebo)	Reduction in anxiety symptoms assessed by using HAMA, STAI	Hedge’s g Fluvoxamine: 1.82 ± 1.63 Placebo: 0.29 ± 2.62	Fluvoxamine superior to a placebo	
Mochcovitch 2010 [35]	SR	Asnis 2001 [61]	8	db	93, 95	Percentage of patients panic-free at the end point	Fluvoxamine: 37% Placebo: 47%	Fluvoxamine inferior to a placebo	Low
		Nair 1996 [68]	8	db	50, 50	Percentage of patients panic-free at the end point	Fluvoxamine: 69% Placebo: 45.7%	Fluvoxamine superior to a placebo	
Du 2021 [38]	NMA	Nair 1996 [68]	8	db	50, 50	Improvement ratio	OR = 1.21 (0.96–1.53)	Fluvoxamine superior to a placebo, but not statistically significantly	Moderate
		Black 1996 [64]	8	db	17, 14				
		Asnis 2001 [61]	8	db	93, 95				
		Sandmann 1998 [72]	6	db	23, 23				
		Sharp 1996 [73]	12	db	29, 28				
Boyer 1995 [34]	MA	Bakish 1993 [62]	NK	db	18, 18	Improvement ratio	OR = 4.75 (4.338–5.154)	Fluvoxamine superior to a placebo	Low
		Black 1993 [63]	8	db	25, 25	Improvement ratio	OR = 3.41 (3.119–3.705)	Fluvoxamine superior to a placebo	
		Westenberg 1989 [75]	NK	db	NK	Improvement ratio	OR = 2.37 (2.087–2.662)	Fluvoxamine superior to a placebo	
		Den Boer 1990 [66]	8	db	20 (fluvoxamine), 20 (ritanserin), 19 (placebo)	Improvement ratio	OR = 5.86 (5.639–xxx)	Fluvoxamine superior to a placebo	
		Hoehn-Saric 1993 [67]	8	db	25, 25	Improvement ratio	OR = 2.30 (1.996–2.608)	Fluvoxamine superior to a placebo	
Fluvoxamine single arm No overlap									
Andrisano 2012 [37]	MA	Pols 1996 [71]	6	sb	11	Reduction in anxiety symptoms assessed by using STAI, ZUNG	Hedge’s g 1.03 ± 2.15	Fluvoxamine significantly improved anxiety symptoms	Moderate
Fluvoxamine vs. Imipramine No overlap									
Mochcovitch 2010 [35]	SR	Nair 1996 [68]	8	db	50, 48	Percentage of patients panic-free at the end point	Fluvoxamine: 37% Imipramine: 64%	Fluvoxamine inferior to Imipramine	Low
Perna 2011 [36]	SR	Perna 2002 [70]	4	db	24 (Fluvoxamine), 27 (Imipramine), 22 (clomipramine), 27 (paroxetine), 23 (sertraline)	Response to treatment assessed by using PASS	Fluvoxamine: 35% Imipramine: 38%	No significant difference	Low
						Reduction in functional impairment and fear symptoms assessed by using SDS, FQ	NK	No significant difference	
Fluvoxamine vs. other drugs No overlap									
Andrisano 2012 [37]	MA	Den Boer 1988 [65]	6	db	20, 24 (Maprotiline)	Reduction in anxiety symptoms assessed by using HAMA	Hedge’s g 1.850 ± 1.03	Fluvoxamine superior to Maprotiline	Moderate
		Van Vliet 1996 [74]	12	db	15, 15 (Brofaromine)	Reduction in anxiety symptoms assessed by using HAMA	Hedge’s g 1.890 ± 1.05	Fluvoxamine superior to Brofaromine	
Perna 2011 [36]	SR	Perna 2002 [70]	4	db	24 (Fluvoxamine), 27 (Imipramine), 22 (clomipramine), 27 (paroxetine), 23 (sertraline)	Response to treatment assessed by using PASS	Fluvoxamine: 35% Sertraline: 75% Clomipramine: 71%	Sertraline and Clomipramine significantly superior to Fluvoxamine	Low
						Reduction of functional impairment and fear symptoms assessed by using SDS, FQ versus sertraline or clomipramine	NK	No significant difference	
		Palatnik 2001 [69]	4	db	10, 11 (Inositol)	Response to treatment versus inositol	Fluvoxamine: 60% Inositol: 72%	No significant difference	

Note: Abbreviations: CAS, Clinical Anxiety Scale; CI, confidence interval; db, double blinded; FQ, the fear questionnaire; HAMA, Hamilton Rating Scale for Anxiety; NK, not known; NMA, network meta-analysis; MA, meta-analysis; MD, mean difference; OR, odds ratio; PASS, Panic-Associated Symptoms Scale; SDS, Sheehan Disability Scale; SMD, standardized mean differences; STAI, State Trait Anxiety Inventory; SR, systematic review; RR, risk ratio; ZUNG, Zung Self-Rating Anxiety Score; xxx—there was a mistake in the number published in the data table.

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
