# Peer review of "The Efficacy of Fluvoxamine in Anxiety Disorders and Obsessive-Compulsive Disorder: An Overview of Systematic Reviews and Meta-Analyses"

_pharmaceuticals, 2025, doi:10.3390/ph18030353_

Round 1
Reviewer 1 Report
Comments and Suggestions for Authors
The manuscript is written well. A minor revision with comments below will improve the article’s quality.
1. The abstract is well-written and a bit lengthy. Authors are suggested to make it more precise.
2. In the abstract, the sentences written under results also seem like methods. The abstract should be restructured.
3. Anxiety and depression are very confusing among common people. Therefore, authors should provide some statements to draw a clear borderline between these two commonest forms of disorder, citing the most recent articles. The authors are advised to cite - https://doi.org/10.3390/ijerph21121620, https://doi.org/10.3390/ph17030366
4. The purpose and novelty of this article are not justified. As Fluvoxamine is an already approved drug, why do the authors think the discussion about it is important without any comparison with other drugs?
5. As the authors stated, ‘fluvoxamine on GAD met our eligibility criteria’, what are the eligibility criteria?
6. What is the exact outcome of this study? The key findings need to be elaborated.
7. Why is the number of studies selected for each disorder different? This way won’t the result be affected or biased?
Author Response
Thank you for the valuable feedback on our manuscript. We have addressed each point in detail below to improve the clarity and quality of our article. All changes made to the manuscript have been highlighted in red in the word version.
- The abstract is well-written and a bit lengthy. Authors are suggested to make it more precise.
Reply: We appreciate this feedback and have revised the abstract to make it more concise while retaining the essential details.
- In the abstract, the sentences written under results also seem like methods. The abstract should be restructured.
Reply: We have restructured the abstract to ensure clear demarcation between the methods and results sections.
The abstract before the changes was: Objective: This systematic review aims to evaluate the efficacy of fluvoxamine in the treatment of anxiety disorders and obsessive-compulsive disorder (OCD) by synthesizing evidence from systematic reviews and meta-analyses. Methods: We conducted a literature search in PubMed and the Cochrane Central Register of Controlled Trials, focusing on fluvoxamine’s efficacy in generalized anxiety disorder (GAD), social anxiety disorder (SAD), panic disorder (PD), and OCD. We included systematic reviews and meta-analyses of randomized controlled trials (RCTs) comparing fluvoxamine to placebo or other drugs. Results: The study included 14 systematic reviews (5 for OCD, 3 for SAD, and 6 for PD), covering 37 RCTs (16 for OCD, 6 for SAD, and 15 for PD), with a total of 3,621 patients (1,745 with OCD, 1,034 with SAD, and 842 with PD). The average treatment duration in the included RCTs was 10.5 weeks for OCD, 13.7 weeks for SAD, and 7.5 weeks for PD. A high-quality systematic review demonstrated that fluvoxamine is superior to placebo in improving symptoms and response rates for OCD, as measured by the Yale-Brown Obsessive-Compulsive Scale (Y-BOCS) and other relevant scales. Three meta-analyses comparing fluvoxamine to clomipramine in OCD found no significant differences in efficacy regarding symptom improvement. However, all these studies were rated as low-quality reviews according to the AMSTAR-2 criteria. Two additional systematic reviews, both rated highly using the AMSTAR-2 tool, confirmed the superiority of fluvoxamine in reducing symptom severity and improving response rates in patients with SAD compared to placebo. However, findings for PD were inconsistent. A Cochrane network meta-analysis, also rated highly by AMSTAR-2, found that while fluvoxamine showed better response rates than placebo, the difference was not statistically significant. Conclusion: Overall, fluvoxamine proved to be effective in the treatment of OCD and SAD. There were no significant differences between fluvoxamine and clomipramine in treating OCD. While some reviews highlighted its potential in alleviating GAD, its impact on panic-specific outcomes remained inconsistent. Clinicians should carefully consider individual patient characteristics and the specific nuances of each disorder when prescribing fluvoxamine.
And after changes the abstract is:
Objective: This systematic review aims to evaluate the efficacy of fluvoxamine in the treatment of anxiety disorders and obsessive-compulsive disorder (OCD) by synthesizing evidence from systematic reviews and meta-analyses. Methods: We conducted a literature search in PubMed and the Cochrane Central Register of Controlled Trials, focusing on fluvoxamine’s efficacy in generalized anxiety disorder (GAD), social anxiety disorder (SAD), panic disorder (PD), and OCD. We included systematic reviews and meta-analyses of randomized controlled trials (RCTs) comparing fluvoxamine to placebo or other drugs. The quality of evidence from the included reviews was assessed using A Measurement Tool to Assess Systematic Reviews - version 2 (AMSTAR-2). Results: The study included 14 systematic reviews (5 for OCD, 3 for SAD, and 6 for PD), covering 37 RCTs (16 for OCD, 6 for SAD, and 15 for PD), with a total of 3,621 patients (1,745 with OCD, 1,034 with SAD, and 842 with PD). A high-quality systematic review demonstrated that fluvoxamine is superior to placebo in improving symptoms and response rates for OCD. Three meta-analyses comparing fluvoxamine to clomipramine in OCD found no significant differences in efficacy regarding symptom improvement. Two additional systematic reviews, both rated as high quality, confirmed the superiority of fluvoxamine in reducing symptom severity and improving response rates in patients with SAD compared to placebo. However, findings for PD were inconsistent. A meta-analysis, also rated as high quality, found that while fluvoxamine showed better response rates than placebo, the difference was not statistically significant. Conclusion: Overall, the efficacy of fluvoxamine in the treatment of OCD and SAD was demonstrated. While some re-views highlighted its potential in alleviating GAD, its impact on panic-specific outcomes remained inconsistent.
- Anxiety and depression are very confusing among common people. Therefore, authors should provide some statements to draw a clear borderline between these two commonest forms of disorder, citing the most recent articles. The authors are advised to cite - https://doi.org/10.3390/ijerph21121620, https://doi.org/10.3390/ph17030366
Reply: We agree with this suggestion and have added a paragraph in the Introduction section to delineate the distinctions between anxiety and depression. We have also cited some additional articles including one of the recommended by the reviewer.
The paragraph and the references added are: Anxiety and depression, though often co-occurring and sharing some overlapping symptoms and treatment options, are distinct psychiatric conditions with unique characteristics (7-9). Anxiety is primarily characterized by excessive fear, worry, and related behavioral disturbances (10). These symptoms are future-oriented and often involve a heightened perception of threat (10). Depression is primarily marked by anhedonia and other symptoms such as feelings of worthlessness, fatigue, and changes in sleep or appetite. In general, symptoms of depression are often past-oriented and linked to feelings of loss or hopelessness (11).
- Zhou X, Lin Z, Yang W, Xiang M, Zhou B, Zou Z. The differences of event-related potential components in patients with comorbid depression and anxiety, depression, or anxiety alone. J Affect Disord. 2023 Nov 1;340:516-522. doi: 10.1016/j.jad.2023.08.049. Epub 2023 Aug 10. PMID: 37572703.
- Choi KW, Kim YK, Jeon HJ. Comorbid Anxiety and Depression: Clinical and Conceptual Consideration and Transdiagnostic Treatment. Adv Exp Med Biol. 2020;1191:219-235. doi: 10.1007/978-981-32-9705-0_14. PMID: 32002932.
- Sartorao, A.L.V.; Sartorao-Filho, C.I. Anxiety and Depression Disorders in Undergraduate Medical Students During the COVID-19 Pandemic: An Integrative Literature Review. Int. J. Environ. Res. Public Health 2024, 21, 1620. https://doi.org/10.3390/ijerph21121620
- Szuhany KL, Simon NM. Anxiety Disorders: A Review. JAMA. 2022;328(24):2431–2445. doi:10.1001/jama.2022.22744
- Han H, Midorikawa A. Depression Accompanied by Hopelessness Is Associated with More Negative Future Thinking. Healthcare (Basel). 2024 Jun 17;12(12):1208. doi: 10.3390/healthcare12121208. PMID: 38921322; PMCID: PMC11204021.
- The purpose and novelty of this article are not justified. As Fluvoxamine is an already approved drug, why do the authors think the discussion about it is important without any comparison with other drugs?
Reply: Although fluvoxamine is an approved and widely utilized drug in clinical practice, there remains uncertainty regarding its efficacy compared to other pharmacological treatments, particularly newer SSRIs that were developed after fluvoxamine, such as sertraline. This uncertainty persists despite fluvoxamine's longstanding use, as the existing body of evidence often lacks robust comparisons or high-quality head-to-head trials. For this reason, our study aimed to go beyond placebo comparisons and specifically explore data involving comparisons between fluvoxamine and other drugs, addressing this critical gap in the literature.
Furthermore, fluvoxamine is unique among SSRIs due to its mechanism of action, which includes sigma-1 receptor agonism. The sigma-1 receptor is known to modulate several neurotransmitter systems, including glutamate and dopamine, and has been associated with anxiolytic, neuroprotective, and antidepressant effects. This distinct pharmacodynamic profile raises the question of whether fluvoxamine might exhibit clinical advantages over other drugs that lack sigma-1 receptor activity. By including evidence on fluvoxamine’s efficacy in comparison to other pharmacological options, our study sought to evaluate whether this mechanism translates into superior clinical outcomes in anxiety and obsessive-compulsive disorders. These insights could be valuable for clinicians seeking to optimize treatment strategies for patients with anxiety disorders and OCD.
- As the authors stated, ‘fluvoxamine on GAD met our eligibility criteria’, what are the eligibility criteria?
Reply: Thank you for your insightful suggestion. To address it, we have revised the sentence and changed as it reads: "After applying the exclusion criteria, no systematic reviews or meta-analyses on the effect of fluvoxamine on GAD were included."
- What is the exact outcome of this study? The key findings need to be elaborated.
Reply: Thank you for your comment. We described the exact outcome of the study in the first paragraph of the discussion section, which provides a detailed synthesis of the key findings. Specifically, this overview of systematic reviews offers a comprehensive assessment of fluvoxamine's efficacy in anxiety disorders and OCD. It highlights that fluvoxamine is effective in treating both OCD and SAD, with significant reductions in symptoms observed for OCD compared to placebo, as confirmed by high-quality systematic reviews. In SAD, fluvoxamine consistently outperformed placebo in reducing symptom severity and improving social functioning. However, its effectiveness in PD is less consistent, with mixed evidence regarding its ability to reduce panic-specific outcomes.
- Why is the number of studies selected for each disorder different? This way won’t the result be affected or biased?
Reply: The variation in the number of studies for each disorder reflects the existing research landscape and availability of systematic reviews. For instance, the effect of fluvoxamine on OCD has been extensively studied, resulting in a greater number of high-quality reviews while SAD and PD have fewer systematic reviews.
We acknowledge that these disparities may introduce bias in the synthesis of results. This limitation has been included on discussion with the follow sentence: We note that disparities in the number of studies for each disorder may bias the results, with less representation for SAD and PD compared to the extensively studied OCD.
Reviewer 2 Report
Comments and Suggestions for Authors
Review on the manuscript of Haddad M et al., (pharmaceuticals-3373851): “The Efficacy of Fluvoxamine in Anxiety Disorders and Obsessive-Compulsive Disorder: An Overview of Systematic Reviews and Meta-analyses”.
In this systematic review, the Authors compiled and analyzed findings from existing systematic reviews and meta-analyses to assess the effectiveness of fluvoxamine in managing anxiety and obsessive-compulsive disorders. The Authors concluded that fluvoxamine is effective in treating obsessive-compulsive disorder and social anxiety disorder. However, fluvoxamine seems not to be more effective than clomipramine for obsessive-compulsive disorder treatment. Additionally, the effectiveness of fluvoxamine in panic disorder were variable.
Overall, I find this topic to be of great interest, as an overview of systematic reviews and meta-analyses offers a high-level synthesis of the available evidence, providing a clearer understanding of fluvoxamine's therapeutic potential in anxiety disorders. Furthermore, a thorough evaluation of fluvoxamine's efficacy in anxiety disorders can help clinicians in making informed decisions and adapting treatment to individual patient needs. I believe the Authors have effectively addressed the primary question posed. Below, I indicate the issues identified in the current version of the manuscript. I hope the Authors find the following comments and suggestions helpful.
1 - In Table 3, the review by Perna (2011) lists the same randomized clinical trial (RCT) (Perna 2002) twice. It might be more streamlined to present the Perna 2002 RCT in a single entry, combining the different outcomes into one line for clarity.
2 - On line 231, the Authors refer to clomipramine as “another SSRI.” However, clomipramine is not an SSRI but rather a tricyclic antidepressant. I kindly suggest that the Authors revise this detail for accuracy.
3 - At the end of the discussion, the Authors suggest that “further large-scale, high-quality RCTs are needed to confirm fluvoxamine’s efficacy, especially in head-to-head comparisons with other SSRIs and tricyclic antidepressants”. While I understand that the Authors aimed to consolidate data from existing reviews and meta-analyses into a single review, I feel that the outcomes of this work are somewhat limited in its current format. A meta-analysis of all RCTs using fluvoxamine in anxiety disorders could provide a much larger included population and generate more robust and valuable insights on this topic.
Minor points:
1 - The title for Table 3 appears to be missing. Could the Authors address this?
Author Response
Thank you for the valuable feedback on our manuscript. We have addressed each point in detail below to improve the clarity and quality of our article. All changes made to the manuscript have been highlighted in red in the word version.
Review on the manuscript of Haddad M et al., (pharmaceuticals-3373851): “The Efficacy of Fluvoxamine in Anxiety Disorders and Obsessive-Compulsive Disorder: An Overview of Systematic Reviews and Meta-analyses”.
In this systematic review, the Authors compiled and analyzed findings from existing systematic reviews and meta-analyses to assess the effectiveness of fluvoxamine in managing anxiety and obsessive-compulsive disorders. The Authors concluded that fluvoxamine is effective in treating obsessive-compulsive disorder and social anxiety disorder. However, fluvoxamine seems not to be more effective than clomipramine for obsessive-compulsive disorder treatment. Additionally, the effectiveness of fluvoxamine in panic disorder were variable.
Overall, I find this topic to be of great interest, as an overview of systematic reviews and meta-analyses offers a high-level synthesis of the available evidence, providing a clearer understanding of fluvoxamine's therapeutic potential in anxiety disorders. Furthermore, a thorough evaluation of fluvoxamine's efficacy in anxiety disorders can help clinicians in making informed decisions and adapting treatment to individual patient needs. I believe the Authors have effectively addressed the primary question posed. Below, I indicate the issues identified in the current version of the manuscript. I hope the Authors find the following comments and suggestions helpful.
1 - In Table 3, the review by Perna (2011) lists the same randomized clinical trial (RCT) (Perna 2002) twice. It might be more streamlined to present the Perna 2002 RCT in a single entry, combining the different outcomes into one line for clarity.
Reply: Thank you for pointing this out. We have revised Table 3 to consolidate the outcomes of the Perna (2002) RCT into a single entry. This modification enhances the clarity and readability of the table.
2 - On line 231, the Authors refer to clomipramine as “another SSRI.” However, clomipramine is not an SSRI but rather a tricyclic antidepressant. I kindly suggest that the Authors revise this detail for accuracy.
Reply: We apologize for this oversight. The manuscript has been corrected to accurately describe clomipramine as a tricyclic antidepressant.
3 - At the end of the discussion, the Authors suggest that “further large-scale, high-quality RCTs are needed to confirm fluvoxamine’s efficacy, especially in head-to-head comparisons with other SSRIs and tricyclic antidepressants”. While I understand that the Authors aimed to consolidate data from existing reviews and meta-analyses into a single review, I feel that the outcomes of this work are somewhat limited in its current format. A meta-analysis of all RCTs using fluvoxamine in anxiety disorders could provide a much larger included population and generate more robust and valuable insights on this topic.
Reply: Thank you for your thoughtful feedback. We appreciate your suggestion regarding the potential value of a meta-analysis of all RCTs using fluvoxamine in anxiety disorders. To address your comment, the mentioned sentence has been removed from the discussion.
Minor points:
1 - The title for Table 3 appears to be missing. Could the Authors address this?
Reply: We apologize for this omission. The title for Table 3 has been added: " Characteristics of included systematic reviews and meta-analyses on the efficacy of fluvoxamine in the treatment of panic disorder."
Round 2
Reviewer 2 Report
Comments and Suggestions for Authors
Second review on the manuscript of Haddad M et al., (pharmaceuticals-3373851): “The Efficacy of Fluvoxamine in Anxiety Disorders and Obsessive-Compulsive Disorder: An Overview of Systematic Reviews and Meta-analyses”.
In this systematic review, the Authors compiled and analyzed findings from existing systematic reviews and meta-analyses to assess the effectiveness of fluvoxamine in managing anxiety and obsessive-compulsive disorders. The Authors concluded that fluvoxamine is effective in treating obsessive-compulsive disorder and social anxiety disorder. However, fluvoxamine seems not to be more effective than clomipramine for obsessive-compulsive disorder treatment. Additionally, the effectiveness of fluvoxamine in panic disorder were variable.
This is the second version of the manuscript following peer review. I acknowledge the Authors for addressing the comments and suggestions raised during the first review round. Below, I leave an issue identified in the current version of the manuscript. I hope the Authors find the following suggestion helpful.
Minor point:
1 - On line 80, please replace “was” by “were”.
Author Response
Thank you for the valuable feedback on our second version of the manuscript. We have addressed each point in detail below to improve the clarity and quality of our article. All changes made to the manuscript have been highlighted in red in the word version.
Reviewer 2
On line 80, please replace “was” by “were”
Thank you for pointing this out and we altered this sentence as suggested.